# A Novel Approach for Assessment of Clonal Hematopoiesis of Indeterminate Potential Using Deep Neural Networks

**Sangeon Ryu**[1]                                                    ALLEN.RYU@YALE.EDU

**Shawn Ahn**[1]                                                      SHAWN.AHN@YALE.EDU

**Jeacy Espinoza**[2]                                                JEACY.ESPINOZA@YALE.EDU

**Alokkumar Jha**[2]                                               ALOKKUMAR.JHA@YALE.EDU

**Stephanie Halene**[3]                                          STEPHANIE.HALENE@YALE.EDU

**James S. Duncan**[1]                                           JAMES.DUNCAN@YALE.EDU

**Jennifer M Kwan\***[2]                                        JENNIFER.KWAN@YALE.EDU

**Nicha C. Dvornek\***[1]                                      NICHA.DVORNEK@YALE.EDU

[1] *Department of Radiology & Biomedical Imaging, Yale School of Medicine, New Haven, USA*

[2] *Section of Cardiovascular Medicine, Yale School of Medicine, New Haven, USA*

[3] *Section of Hematology, Yale School of Medicine, New Haven, USA*

*\*co-corresponding authors*

**Editors:** Under Review for MIDL 2023

## Abstract

We propose a novel diagnostic method for clonal hematopoiesis of indeterminate potential (CHIP), a condition characterized by the presence of somatic mutations in hematopoietic stem cells without detectable hematologic malignancy, using deep-learning techniques. We developed a convolutional neural network (CNN) to predict CHIP status using 4 different views from standard delayed gadolinium-enhanced cardiac MRI. We used 5-fold cross validation on 82 patients to assess the performance of our model. Different algorithms were compared to find the optimal patient-level prediction method using the image-level CNN predictions. We found that the best model had an AUC of 0.85 and an accuracy of 82%. We conclude that a deep learning-based diagnostic approach for CHIP is promising.

**Keywords:** Deep learning, clonal hematopoiesis of indeterminate potential, cardiovascular disease, cardiac MRI

## 1. Introduction

Clonal hematopoiesis of indeterminate potential (CHIP) is an age-related premalignant condition, characterized by the presence of clonally expanded hematopoietic stem cells caused by a leukemogenic mutation in individuals without evidence of hematologic malignancy (Marnell et al., 2021). CHIP is an independent risk factor for cardiovascular diseases (CVDs), such as atherosclerosis, myocardial infarction, and congestive heart failure (Mooney et al., 2021). CVDs such as these are one of the leading causes of morbidity and mortality worldwide; thus, being able to augment the identification of CHIP beyond DNA sequencing is imperative. Further, although CHIP independently increases the risk of heart disease and heart failure, not all CHIP patients develop these adverse cardiovascular events. Thus, use of machine learning approaches can potentially identify imaging features that can risk stratify who may develop CVD amongst CHIP patients.

RYU AHN ESPINOZA JHA HALENE DUNCAN KWAN* DVORNEK*

Traditionally, CHIP is diagnosed through next-generation sequencing (NGS), a technique that can determine a person's DNA sequence. For this, however, the patient's blood or bone marrow sample must be acquired, almost always through invasive means. As NGS can take hours to days to return a result as well, a quicker, non-invasive method for evaluation of CHIP becomes more desirable.

Preliminary data shows that CHIP is associated with increased fibrosis in human engineered heart tissue. Delayed gadolinium enhancement (DGE) is the method of choice for detecting myocardial fibrosis in magnetic resonance imaging (MRI). Thus, we sought to explore whether fibrosis burden and fibrosis features on cardiac MRI (cMRI) via DGE signatures could indicate if the patient had CHIP.

## 2. Methods

We enrolled an anonymized collection of DGE-cMRI images from 82 patients (42% with CHIP), whose genomic DNA was extracted from peripheral blood samples and sequenced to determine CHIP. Cardiac MRI was performed on 1.5 and 3T scanners, with DGE evaluation performed 8-10 minutes after administration of contrast. Each patient had up to 4 different views in their collection of cMRIs: short-axis view (SAS); 4-chamber view (4CH); vertical long axis (VLA); and left ventricular outflow view (LVOT). Multiple views were incorporated into the prediction model so that we could capture a more complete overview of the heart. Each patient had up to 5-7 SAS views for their cMRIs, but only one cMRI image for the other 3 views; some patients had fewer SAS images and/or were missing one or more of the other 3 views. Missing views were replaced by images with all 0s.

The model was a CNN designed for binary classification (Fig. 1), with 4 convolutional layers and 3 max pooling layers. Uniquely, we incorporated all 4 views as inputs to the model. Each view underwent processing by the convolutional layers, and features from the final convolutional layer were concatenated and processed by fully connected layers. The output of the model gave the probability of CHIP based on the 4-view cMRI sample.

The model was trained using a 5-fold cross validation framework in order to assess the performance of the model. Each fold contained between 14 and 15 patients; we ensured that all cMRI images belonging to the same patient were in the same fold. Each of the 5 folds were used once as a test set, while the other 4 folds were combined to be the training set. In addition to standard image data augmentation techniques, as SAS view included multiple image slices, random combinations of the 4 views were used to augment the number of samples. The model for each fold was trained using binary cross-entropy loss for 300 epochs and then evaluated on the test set. The evaluation profiles were then combined to give an overview of the model architecture's performance in the binary classification task using receiver operating characteristic (ROC) curve analysis.

The model was classifying on an "image-level" - that is, it was classifying each of the image sets (one cMRI from each of the 4 views) into one of the two categories, "CHIP" or "NO CHIP". To extend this to the "patient-level" - that is, combining the predictions for all the images belonging to a patient to make a single classification for the patient themselves - we tested different thresholding approaches for combining the image-level predictions to make a prediction for the patient. Specifically, two approaches were explored: the ratio thresholding method, which took the portion of the image sets belonging to a patient that

were classified in the CHIP category, and if the ratio was greater than the threshold (=0.4), the patient was classified as CHIP; and the max thresholding method, which classified a patient as CHIP if the patient's 4-view image set with maximum probability of CHIP was greater than the threshold.

## 3. Results and Conclusions

We found that between the two thresholding methods, the ratio-thresholding approach performed much better than the max-thresholding method (AUC=0.85 vs. AUC=0.63, Fig. 2). In addition, using the ratio-thresholding method, our approach was able to predict the patient's CHIP status with an accuracy of 82%.

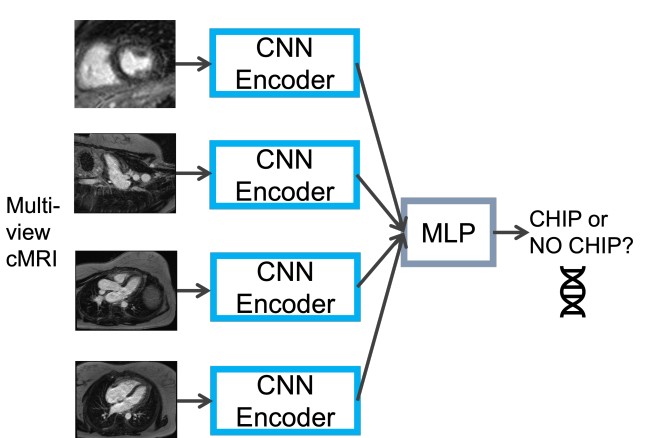

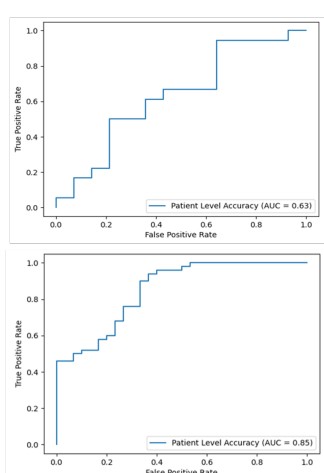

Figure 1: Network architecture for CHIP classification from multi-view DGE-cMRI. CNN, convolutional neural network; MLP, multilayer perceptron.

Figure 2: ROC curves of the two thresholding methods. Top=max-thresholding, bottom=ratio-thresholding.

In conclusion, we proposed a novel approach for determining CHIP from multi-view DGE-cMRI. Our promising early results suggest non-invasive, routine imaging may supplement the diagnosis of CHIP. Future work will extend validation of our approach on large public datasets (e.g., TOPMed) and apply model interpretation techniques (Adebayo et al., 2018) to identify cMRI biomarkers for CHIP as well as imaging features that can predict adverse cardiovascular outcomes in CHIP patients.

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
