# OpenReview forum: "A Novel Approach for Assessment of Clonal Hematopoiesis of Indeterminate Potential Using Deep Neural Networks"
_MIDL.io/2023/Short_Paper_Track — MIDL 2023 Short paper track Poster_

### Official Review · Reviewer_i6eC · 2023-04-23
**The proposed network is able to help diagnosing CHIP but the overall structure is too simple.**

**Rating:** 7
**Confidence:** 4

**Review:**

This paper proposed a new net work to combine 4 views of cMRI and make diagnosis of CHIP, but the overall structure of network is too simple.

---

### Official Review · Reviewer_ufZC · 2023-04-24
**A Novel Approach for Assessment of Clonal Hematopoiesis of Indeterminate Potential Using Deep Neural Networks**

**Rating:** 6
**Confidence:** 3

**Review:**

This paper proposes a deep learning approach to predict the presence of clonal hematopoiesis of indeterminate potential (CHIP) from MRI images, with a ground truth confirmed by DNA sequencing.

PROS
This work seems like a proof-of-concept for a novel application, although not clearly stated as such by the authors.
Ways to combine predictions when multiple images are available are investigated, resulting in realists ranging from AUC=0.63 to AUC=0.85.

CONS
Only 5-fold cross validation is used, so an external validation would be needed to confirm the promising results of this work.
The used architecture is fairly simple and custom for this application, also using 2D data instead of 3D, I wonder if using deeper architectures and/or a 3D approach would not lead to better performance, and why a multi-view 2D approach was preferred here.